# OpenReview forum: "Shape it Up! Restoring LLM Safety during Finetuning"
_NeurIPS.cc/2025/Conference — NeurIPS 2025 poster_

### Official Review · Reviewer_p3z3 · 2025-07-01

**Clarity:** 3
**Significance:** 3
**Originality:** 2
**Rating:** 4
**Confidence:** 4

**Summary:**

This paper introduces dynamic safety shaping (DSS), a fine-tuning framework for large language models that leverages fine-grained safety signals to reinforce safe content and suppress unsafe segments within model responses. By repurposing guardrail models to evaluate safety at the segment level, the authors propose Safety Trajectory Assessment of Response (STAR), a token-level signal enabling dynamic safety control during training. Experiments show that the proposed method achieves substantial safety gains across models and datasets without sacrificing performance on intended tasks.

**Questions:**

See cons.

**Ethical Concerns:**

["NO or VERY MINOR ethics concerns only"]

**Final Justification:**

The author's feedback answers most of my concerns, so I decided to keep the ratings.

**Limitations:**

No. The paper needs to discuss the limitations.

**Quality:**

3

**Strengths And Weaknesses:**

Pros:

- 1 This paper studies an important problem in LLM safety. It formalises the limitations of “static” sample-level shaping (e.g., RS) and reframes safety mitigation as a sequence-level weighting problem.
- 2  The method is evaluated across multiple guardrails and benchmark datasets,  outperforming a few baseline methods in Safety metrics.
- 3  The theoretical bound (Theorem 1) is provided. It ties post-finetuning harmfulness to the expected sequence-level KL divergence.

Cons:
- 1 The key idea of this paper is similar to Deep token, making the contribution of this paper a bit of limited.
- 2 The proposed method is heavily dependent on guardrail quality. Safety still degrades almost linearly with the guard-rail FN rate, even though the decline is gentler than RS, DSS cannot handle entirely novel jailbreak styles unseen by the guard-rail.
- 3 Code and data release are “upon acceptance”; check-points and guardrail configs are not yet public, which may slow reproducibility.
- 4 Only short-answer benchmarks (MMLU, ARC-C, GSM-8K) are used; no tests on generation-heavy tasks (long-form reasoning, coding) where the KL term might over-regularise helpful content.

---

> ### Author Rebuttal · Authors · 2025-07-31
>
> We are grateful to the reviewer for acknowledging the significance of our findings and contributions! We are glad that the reviewers highlight the strengths of our work:
> - STAR-DSS is a **novel, fine-grained approach to robustly mitigate LLM finetuning risks** by dynamically shaping updates using token-level safety signals (qGV6, uk2R, wxoy, p3z3).
> - It is **theoretically grounded**, introducing the first upper bound on fine-tuned LLM safety via KL divergence and STAR scores (qGV6, wxoy, p3z3).
> - It demonstrates strong and generalizable empirical results across diverse models, guardrails, and threat scenarios, **achieving safety gains without compromising task performance** (qGV6, uk2R, wxoy, p3z3).
>
> We hope our responses below address your specific concerns.
>
> > Significant differences between our STAR-DSS and Deep Token
>
> While both methods apply token-level shaping, our approach significantly advances over Deep Token in both **theoretical grounding** and **empirical robustness**.
>
> **Theoretical advances.**
> As detailed in Appendix E, Deep Token interpolates between cross-entropy (CE) and KL loss using a fixed, handcrafted token-wise schedule $\{\beta_t\}$, typically set as $\left( \beta_1, \beta_{2:5}, \beta_{>5} \right) = \left(0.2, 1.0, 0.1 \right)$, assuming early tokens are more safety-critical. This position-based heuristic ignores the actual safety context. In contrast, **STAR-DSS replaces hand-crafted schedules with a token-level safety signal — the STAR score — dynamically derived from guardrail model logits**. Given a prefix $\left(x, y_{1:t} \right)$, we compute $\text{STAR}^{(t)}$ and use it to modulate the loss:
>
> - If $\text{STAR}^{(t)} \approx 1$: the token is safe → CE dominates ($\beta_t \to 0$)
> - If $\text{STAR}^{(t)} \approx 0$: the token is unsafe → KL dominates ($\beta_t \to \infty$)
>
> This leads to three key advantages:
>
> - **Data-driven adaptivity**: STAR-DSS responds to the actual evolving safety risk within a response, rather than relying on position-based heuristics.
> - **No manual tuning**: Only a single global hyperparameter $\lambda_{\text{KL}}$ is needed to balance CE and KL terms — eliminating token-level tuning.
> - **Theoretical guarantee**: As shown in Sec. 5.2, STAR-DSS satisfies an upper bound on model harmfulness, tied to the guardrail’s FN rate and KL divergence from the reference model. Deep Token does not provide such guarantees.
>
> **Empirical robustness.**
> We reproduce Deep Token using their official code and run both methods in the same worst-case setup (PureBad finetuning, no safe data), matching Deep Token’s experimental protocol (Sec. 4.2 in their paper). As shown in Table A, performance is close on Llama-2-7B-Chat (the model used in Deep Token) — though STAR-DSS still improves safety over Deep Token. However, when applied to a newer model (Llama-3.2), Deep Token’s performance degrades significantly, confirming that its fixed schedule does not generalize across models. In contrast, STAR-DSS remains robust across model families and sizes.
>
> **Table A: Comparison with Deep Token under the worst-case setting. Our method adapts seamlessly without parameter tuning and matches the original model’s safety.**
>
> | Model           | Method       | HEx-PHI (%) | AdvBench (%) |
> | :----------| :--------------|------------: |-------------: |
> | Llama-2-7B-Chat       | Deep Token   | 86.67       | 95.38         |
> |                          | STAR-DSS     | 96.36       | 99.81         |
> | Llama-3.2-1B-Instruct | Deep Token   | 35.76       | 51.54         |
> |                          | STAR-DSS     | 72.12       | 89.42         |
>
> ---
>
> Overall, STAR-DSS generalizes Deep Token’s approach with a **principled, guardrail-driven formulation**. It is more adaptive, less brittle across models, and comes with theoretical safety guarantees, making it a stronger and more general solution for finetuning-time safety alignment.
>
> > Dependence on guardrail quality and generalization to novel attacks
>
> **Robustness to the guardrail quality**
> We acknowledge that the upper bound on model harmfulness scales with the guardrail’s false negative (FN) rate, as shown in our theoretical analysis (Appendix C). However, **STAR-DSS remains robust in practice**, maintaining strong safety even with weaker guardrails. As shown in Table B (also in Fig. 4b and Table 13), safety improves with lower FN rates, yet STAR-DSS still performs well when moving from Granite Guardian (3% FN) to Llama Guard (15–18% FN) — a 5–6× increase — with only modest drops in safety on HEx-PHI and AdvBench.
>
> **Table B: STAR-DSS maintains strong safety and capability across four guardrail models from two families. All results are reported under the worst-case scenario.**
>
> | Guardrail               | FN on PureBad (%) | HEx-PHI (%) | AdvBench (%) |
> |:---------------|---------------:|-------------:|---------------:|
> | Granite Guardian-3.1-2B | 3                 | 72.12       | 89.42         |
> | Granite Guardian-3.1-8B | 3                 | 75.45       | 90.19         |
> | Llama Guard-3-1B           | 15                | 69.09       | 87.88         |
> | Llama Guard-3-8B          | 18                | 72.12       | 85.58         |
>
> ---
>
> **Robustness under adversarial adaptation.**
> To assess robustness under adversarial adaptation, we conduct a response adaptation attack by appending misleading suffixes to all PureBad responses, causing Granite Guardian 3.1-2B’s FN rate to spike from 3% to 34% (Appendix G, Table 7). This severely impacts vanilla SFT and RS, which rely on full-sequence acceptance: once the suffix fools the guardrail, harmful content is learned. In contrast, **STAR-DSS remains robust, achieving significantly higher safety scores**. This is because STAR-DSS applies token-level shaping, evaluating the evolving safety of partial completions. Even if a suffix misleads the final classifier, STAR scores for earlier harmful segments remain low, triggering KL regularization and preventing unsafe updates. As shown in Table C, STAR-DSS substantially outperforms both baselines in safety, with no degradation in task performance.
>
> **Table C: Response adaptation attack with a misleading suffix. This attack appends a misleading suffix to
> harmful completions, tricking guardrail models into classifying them as safe. RS, which relies on binary filtering,
> fails to exclude these examples, allowing unsafe data to enter training. STAR-DSS remains robust by using token-level
> STAR scores to suppress unsafe content even after the suffix is introduced.**
>
> | Method         | HEx-PHI (%) | AdvBench (%) | MMLU (%) | ARC-C (%) |
> | :-----------|-------------:|---------------:|-----------:|------------:|
> | Vanilla SFT        | 3.63        | 1.35          | 46.91     | 58.63      |
> | Rejection Sampling (RS) | 3.62    | 14.81         | 47.13     | 58.71      |
> | STAR-DSS (Ours) | 70.91       | 84.62         | 47.25     | 59.23      |
>
> ---
>
> While no method can guarantee robustness to all future jailbreaks, STAR-DSS directly benefits from ongoing improvements to guardrail models. As new threats are identified and integrated into guardrail training, STAR-DSS becomes even stronger — making it a flexible and forward-compatible defense.
>
> > Release code, data, and checkpoints
>
> We fully agree on the importance of reproducibility. At submission time, our institutional legal review was still ongoing, which prevented us from releasing assets. That process is now complete, and **all code, data, and checkpoints are approved for public release**. While we are unable to share links during the rebuttal phase per policy, we are happy to provide an anonymous repository if permitted by the AC and preferred by the reviewer.
>
> > More results on generation-heavy tasks to demonstrate that KL term does not over-regularize
>
> To assess STAR-DSS on generation-heavy tasks, we evaluate on MATH500 [1], a benchmark of 500 diverse math problems requiring long-form reasoning across topics like probability, algebra, and geometry. As shown in Table D, STAR-DSS performs on par with vanilla SFT, showing no evidence of over-regularization from the KL term in our loss formulation.
>
> **Table D: STAR-DSS maintains performance on MATH500, a long-form reasoning benchmark, demonstrating that the KL term does not over-regularize generation.**
>
> |  | Vanilla SFT  | STAR-DSS |
> | :-- | --: | --: |
> | Math500 | 11.6 | 11.8 |
>
>
> [1] Let's Verify Step by Step

---

### Official Review · Reviewer_wxoy · 2025-07-01

**Clarity:** 3
**Significance:** 4
**Originality:** 3
**Rating:** 5
**Confidence:** 4

**Summary:**

The paper proposes Dynamic Safety Shaping (DSS), a novel framework for maintaining the safety of large language models (LLMs) during fine-tuning. Traditional static methods like rejection sampling treat entire training examples as safe or unsafe, overlooking harmful content embedded within otherwise benign data. To address this, the authors introduce the STAR (Safety Trajectory Assessment of Response) score—a token-level signal derived from guardrail models that estimates the evolving safety risk throughout a response. Leveraging this signal, they design STAR-DSS, a dynamic loss function that adaptively balances imitation learning and safety regularization at the token level. The paper provides both theoretical guarantees and extensive empirical evaluations, demonstrating that STAR-DSS successfully maintains model safety across various threats and models without sacrificing task performance.

**Questions:**

1. The proposed method relies heavily on guardrail accuracy. Can the authors provide a more detailed analysis (e.g., ablation or sensitivity study) on how STAR-DSS behaves when guardrail false negatives increase or shift under adversarial adaptation?
2. As shown in Figure 4(b), could you provide some analysis on Llama Guard-3 1B is better than 8B?

**Ethical Concerns:**

["NO or VERY MINOR ethics concerns only"]

**Final Justification:**

The authors addressed my concerns, and I think it's a good job of avoiding breaking the safety mechanism, although it may introduce additional computational cost.

**Limitations:**

Discussed in weaknesses.

**Paper Formatting Concerns:**

1. Figure 5(b) needs to be adjusted.
2. The checklist needs to be rechecked. (e.g. some experimental setting/details are listed in appendix Table 8 not only in Section 6).
3. Limitations should be discussed in detail.

**Quality:**

4

**Strengths And Weaknesses:**

Strengths:
1. The paper proposes a dynamic loss function that **adjusts the loss across different tokens/chunks using STAR scores**. This fine-grained shaping significantly reduces the risk of safety degradation during finetuning compared to static methods.
2. The experiments span multiple LLMs, datasets, harm levels, and guardrails. This comprehensive setup strengthens the generalizability and credibility of the results.
3. STAR-DSS consistently outperforms baselines in both worst-case and ideal-case settings, achieving high safety without sacrificing task performance.

Weaknesses:
1. The interplay between STAR granularity (chunk size M) and computational cost vs. safety-performance trade-off is acknowledged but not deeply optimized or ablated.
2. More discussion on limitations is needed, such as time cost (**total** time of fine-tuning using STAR-DSS vs. baseline), requirements on the Guardrail model (intuitively, the Guardrail model should usually be of similar size to the fine-tuned model to ensure adequate safety).

---

> ### Author Rebuttal · Authors · 2025-07-31
>
> We are grateful to the reviewer for acknowledging the significance of our findings and contributions! We are glad that the reviewers highlight the strengths of our work:
> - STAR-DSS is a **novel, fine-grained approach to robustly mitigate LLM finetuning risks** by dynamically shaping updates using token-level safety signals (qGV6, uk2R, wxoy, p3z3).
> - It is **theoretically grounded**, introducing the first upper bound on fine-tuned LLM safety via KL divergence and STAR scores (qGV6, wxoy, p3z3).
> - It demonstrates strong and generalizable empirical results across diverse models, guardrails, and threat scenarios, **achieving safety gains without compromising task performance** (qGV6, uk2R, wxoy, p3z3).
>
> We hope our responses below address your specific concerns.
>
> > Ablation on STAR Granularity (chunk size $M$) vs. Computational Cost and Safety
>
> We ablate STAR chunk size $M$ to quantify its impact on safety and efficiency. **Smaller $M$ yields finer-grained STAR scores, improving safety at the cost of modest preprocessing overhead**. As shown in Table A, reducing $M$ consistently improves performance on HEx-PHI and AdvBench. Table B reports wall-clock preprocessing time per prompt across models and datasets. We select $M = 5$ as a practical trade-off that achieves strong safety gains with reasonable compute.
>
> **Table A: Smaller chunk size $M$ improves safety by enabling finer-grained STAR scores (mean ± std over 5 runs). LLM: Llama-3.2-1B-Instruct, Guardrail: Granite Guardian-3.1-2B.**
>
> | Safety Benchmark | M=1 | M=5 | M=10 | M=15 |
> | :-- |  --: |  --: |  --: |  --: |
> | HEx-PHI | 76.67±1.2 | 72.22±0.8 | 71.05±1.0 |  70.01±1.3 |
> | AdvBench | 93.08±1.0 | 90.59±1.1 | 88.08±0.9 | 87.00±0.4 |
>
> ---
>
> **Table B: Preprocessing time (sec/prompt) for STAR scoring at different $M$. Experiments were conducted on a single node with 8 A40 GPUs with batch size 100.**
>
> | Guardrail | Dataset | M = 1 | M = 5 | M = 10 | M = 15 |
> | :-- |  :-- |  --: |  --: |  --: |  --: |
> | Granite Guardian-3.1-2B  | PureBad | 1.15 ± 0.01     | 0.28 ± 0.01      | 0.24 ± 0.01      | 0.21 ± 0.00      |
> | Granite Guardian-3.1-2B  | GSM8K   | 0.72 ± 0.00   | 0.15 ± 0.01      | 0.08 ± 0.01      | 0.05 ± 0.00      |
> | Granite Guardian-3.1-8B  | PureBad | 2.77 ± 0.01    | 0.82 ± 0.01      | 0.62 ± 0.01      | 0.55 ± 0.00      |
> | Granite Guardian-3.1-8B | GSM8K   | 2.00 ± 0.01   | 0.40 ± 0.00      | 0.20 ± 0.01      | 0.14 ± 0.00      |
>
> > More discussion on limitations, including computational cost and guardrail model requirements
>
> **Computational cost.** As shown in Table B, computing STAR scores introduces additional preprocessing time. However, this overhead can be mitigated in practice: STAR scores are precomputed once and cached alongside token positions, enabling efficient per-token weighting during training with minimal runtime cost. Furthermore, this scoring step can be parallelized with model training in a pipelined fashion — similar to pipeline parallelism — where STAR score computation overlaps with supervised finetuning on previous batches. We will clarify this efficiency strategy in the revised version.
>
> **Guardrail model requirements.** Our method is agnostic to the specific choice of guardrail model. While a lower FN rate typically leads to stronger STAR-DSS performance (as shown by our theoretical bound in Appendix C), this does not require a large model. For example, on the GuardBench leaderboard [1], IBM Granite Guardian-3.1-2B outperforms larger models like Llama Guard-3-8B and Google ShieldGemma-9B on standard safety benchmarks. This allows practitioners to use compact, efficient guardrails tailored to their alignment needs. Importantly, different guardrails encode different safety policies [2], so the best choice depends on domain and goals — not necessarily model size.
>
> > Sensitivity to guardrail false negatives (FN) and adversarial adaptation
>
> **Robustness to the guardrail quality**
> We acknowledge that model harmfulness is theoretically bounded by the guardrail’s FN rate (Appendix C). However, **STAR-DSS remains robust in practice**, maintaining strong safety even with weaker guardrails. As shown in Table C (also in Fig. 4b and Table 13), safety improves with lower FN rates, yet STAR-DSS still performs well when moving from Granite Guardian (3% FN) to Llama Guard (15–18% FN) — a 5–6× increase — with only modest drops in safety on HEx-PHI and AdvBench.
>
> **Table C: STAR-DSS maintains strong safety and capability across four guardrail models from two families. All results are reported under the worst-case scenario.**
>
> | Guardrail               | FN on PureBad (%) | HEx-PHI (%) | AdvBench (%) |
> |:---------------|---------------:|-------------:|---------------:|
> | Granite Guardian-3.1-2B | 3                 | 72.12       | 89.42         |
> | Granite Guardian-3.1-8B | 3                 | 75.45       | 90.19         |
> | Llama Guard-3-1B           | 15                | 69.09       | 87.88         |
> | Llama Guard-3-8B          | 18                | 72.12       | 85.58         |
>
> ---
>
> **Robustness to adversarial adaptation.**
> To assess robustness under adversarial adaptation, we conduct a response adaptation attack by appending misleading suffixes to all PureBad responses, causing Granite Guardian 3.1-2B’s FN rate to spike from 3% to 34% (Appendix G, Table 7). This severely impacts vanilla SFT and RS, which rely on full-sequence acceptance: once the suffix fools the guardrail, harmful content is learned. In contrast, **STAR-DSS remains robust, achieving significantly higher safety scores**. This is because STAR-DSS applies token-level shaping, evaluating the evolving safety of partial completions. Even if a suffix misleads the final classifier, STAR scores for earlier harmful segments remain low, triggering KL regularization and preventing unsafe updates. As shown in Table D, STAR-DSS substantially outperforms both baselines in safety, with no degradation in task performance.
>
> **Table D: Response adaptation attack with a misleading suffix. This attack appends a misleading suffix to
> harmful completions, tricking guardrail models into classifying them as safe. RS, which relies on binary filtering,
> fails to exclude these examples, allowing unsafe data to enter training. STAR-DSS remains robust by using the token-level
> STAR scores to suppress unsafe content even after the suffix is introduced.**
>
> | Method         | HEx-PHI (%) | AdvBench (%) | MMLU (%) | ARC-C (%) |
> | :-----------|-------------:|---------------:|-----------:|------------:|
> | Vanilla SFT        | 3.63        | 1.35          | 46.91     | 58.63      |
> | Rejection Sampling (RS) | 3.62    | 14.81         | 47.13     | 58.71      |
> | STAR-DSS (Ours) | 70.91       | 84.62         | 47.25     | 59.23      |
>
> > Analysis of Llama Guard-3 1B vs. 8B in Figure 4(b)
>
> In Fig. 4(b), Llama Guard-3 1B slightly outperforms the 8B variant on AdvBench. However, the reverse holds for HEx-PHI (Table C), where the 8B model performs better. This reflects **differing emphases across benchmarks and evolving safety policies between model versions.** As highlighted in a recent survey [2], Llama Guard-3 1B performs better on general regulation tasks, while the 8B excels in domains like social media and cybersecurity — likely due to updated training data or policy shifts. These differences can lead to non-monotonic safety behavior even within the same model family.
>
> While STAR-DSS is guardrail-agnostic and works across families and sizes, we recommend selecting a guardrail tuned to the target safety domain, ideally with low false negative rates in that space.
>
> > Figure 5(b) needs to be adjusted.
>
> Thank you for pointing this out. The issue with Fig. 5(b) was due to a rendering bug that affects certain PDF viewers. We have fixed this in the Appendix Fig. 8.
>
> > The checklist needs to be rechecked. (e.g. some experimental setting/details are listed in appendix Table 8 not only in Section 6).
>
> Thank you for catching this. We will update Checklist Item 6 in the final version to explicitly reference Appendix Table 8 & Section I, in addition to Section 6, as they jointly provide the full experimental details.
>
> [1] GuardBench: A Large-Scale Benchmark for Guardrail Models.
> [2] GUARDSET-X: Multi-Domain, Policy-Grounded, AI Security Guardrail Benchmark.

---

> > ### Comment · Reviewer_wxoy · 2025-08-05
> >
> > Thank you for your reply, my concerns have been resolved.

---

> > > ### Author Response · Authors · 2025-08-05
> > >
> > > Thank you for your thoughtful feedback and for engaging with us during the rebuttal process. We're glad to hear that your concerns have been resolved. We will incorporate the additional experiments and clarifications discussed into the final version of the paper.

---

### Official Review · Reviewer_uk2R · 2025-07-02

**Clarity:** 3
**Significance:** 2
**Originality:** 2
**Rating:** 5
**Confidence:** 3

**Summary:**

This paper is about a novel loss for fine-tuning models via SFT, that take into account the safety of the completions at the token level and thus limit the degradation of model safety performance. The authors place their work specifically in the context of fine-tuning platforms which are available online. The authors introduce the STAR score: given a chunk of text and a safety classifier, the STAR score measures the probability that it’s safe (via the ratio of the logit for “safe” and that for “safe” + that for “unsafe”). The novel loss - DSS for dynamic safety shaping - is a weighted cross-entropy loss with the STAR score,  balanced with the KL divergence to the original model, with a weight of 1- STAR. The authors proceed to demonstrate a theoretical result that bounds the probability for the fine-tuned model to produce unsafe completions with that of the original model, a term linked to the KL divergence of the two models, and the false negative rate of the classifier. Finally, the authors provide experimental results in the following settings
* “Vanilla” SFT or “malicious” user SFT - meaning the input dataset is 100% unsafe;
* “Benign” user SFT - the user has a 5% unsafe dataset;
* “Sophisticated malicious” user SFT - the user has appended misleading prefixes or suffixes to all the completions which are otherwise 100% unsafe, or is conducting a prompt poisoning attack, or trying to do harmful prefilling.

**Questions:**

See strengths and weaknesses above

**Ethical Concerns:**

["NO or VERY MINOR ethics concerns only"]

**Final Justification:**

The authors have replied to my concerns. Their additional experiments (100% benign data + DSS, fine-tuning and testing on the same dataset in various situations re percentage of unsafe training data), the addition of the standard deviations, the discussion around DSS-star limitations (compute costs, vulnerability to targeted attacks) and additional citations should figure in the paper (core or appendix) as to make clearer the benefits of the approach and its limits.

**Limitations:**

The paper is missing a study of when DSS fails (besides the safety classifier performing poorly). In which conditions does the method break? Are there attacks that it cannot help fight against?

**Quality:**

2

**Strengths And Weaknesses:**

**Major comments**

It’s great that the authors are showing results w/ a few safety classifiers, input checkpoints, fine-tuning datasets. However, experimental results are not convincing for a few reasonsM
* There are no standard deviations - e.g. over eval re-runs (the easier). Safety results are noisy, and in the case at hand, results from the proposed method are close to that of rejection sampling, which is likely to be much faster and more compute-efficient.
* In relation to this, no data is presented regarding time and cost of a run w/ DSS. It would be interesting to see how that changes w/ M (chunk size). What's the tradeoff exactly if someone decided to use the DSS rather than rejection sampling?
* I don’t find the “vanilla” SFT scenario very credible. Providers have access to safety classifiers too, and it’s much simpler to plan for pausing someone’s account once a certain threshold of harmful content is passed, to investigate, rather than force them to use another loss, that’d additionally incur the user or them more cost.
* There should be more references to other literature mentioning prefix/suffix appending type of attacks, as otherwise they seem a little anecdotal (“response adaptation attack” l. 355). More generally, there should be more coverage of the background. It seems from most of the paper that benign fine-tuning needs to include unsafe examples to deteriorate the safety performance (benign user SFT scenario), but it’s not the case. It would be interesting to see how the DSS fares in that case (see minor comment below for a suggestion of where to add it).

Overall, I find that the most convincing application of the authors’ method is to defend against data poisoning attacks and harmful prefilling attacks, which are mentioned in section 6.4. In that case, yes the model fine-tuner themselves are looking for solutions to mitigate against someone else’s attacks. It’s likely that depending on the user's application they’d be willing to incur more cost / longer training time to make sure that even if their data was poisoned, backdoors could not be activated. I think it could worth it to focus experimental results more on these aspects in a future version of the paper.

**Minor comments**
* Example in figure 1 is not super convincing. “Here is” is not essentially safe, and “creating scripts for exploiting vulnerabilities” isn’t essentially unsafe either. I’d suggest to find a few examples that are more illustrative and display them.
* The baseline is missing in table 2 - what are the results if the user fine-tunes on their corrupted dataset? What about if they fine-tune on GSM8K alone? The baseline would also be a great addition to figure 4.c, keeping the same colour for it over all subplots in Figure 4 and explaining it in the figure caption (its color changing from blue in 4.a to grey in 4.d).
* It’s surprising that the authors are presenting results that involve fine-tuning on one dataset (eg PureBad), and safety results on other ones (HexPhi and AdvBench). Often, one wants to fine-tune on the same data distribution that they want to apply the model to.
* Fig. 5b is hard to read.

---

> ### Author Rebuttal · Authors · 2025-07-31
>
> We thank the reviewer for all the constructive suggestions. We begin by clarifying the motivation and threat model, as several concerns appear to arise from misunderstandings of our intended setting. Detailed responses follow.
>
> **Threat Model and Motivation.**
> Our work targets realistic risks in fine-tuning-as-a-service platforms, where users can upload data that directly updates model weights. This opens the door to harmful fine-tuning attacks, as shown by Qi et al. [2], who compromise GPT-3.5 Turbo with just 10 harmful examples ($<0.20), and recent work [3] that achieves similar success on GPT-4 — despite platform safeguards.
>
> Even benign users may unknowingly fine-tune on unsafe content (e.g., from open-source datasets downloaded from HuggingFace). If moderation misses these examples, safety alignment degrades severly. STAR-DSS is designed to mitigate this risk, whether harm is intentional or accidental.
>
> **Experimental Design Clarification.**
> We evaluate STAR-DSS across six realistic scenarios, varying:
> - Whether user data is benign, partially unsafe, or entirely harmful
> - Whether the provider supplements training with a trusted safe dataset
>
> While clean data (e.g., GSM8K) rarely affects safety [4, 5], even small amounts of harmful data cause significant degradation. We focus on these high-risk cases (Cases 3–6), while including benign baselines (Cases 1–2) for comparison.
>
> **Table A: Evaluated fine-tuning scenarios by user intent and provider safeguards**
>
> |   | Purely Benign | 5% Unsafe Mixed In  | Purely Harmful |
> | :--- | :--- | :--- | :----|
> | **No Safety Data**  | Case 1: Benign SFT (baseline)→ Table F below | Case 3: Mixed SFT→ Table 1 (main paper) | Case 5: Malicious SFT→ Table 10 (appendix) |
> | **With Safety Data**| Case 2: Benign SFT + $D_{\text{safe}}$→ Table F below | Case 4: Mixed + $D_{\text{safe}}$→ Table 9 (appendix) | Case 6: Malicious + $D_{\text{safe}}$→ Table 2 (main paper) |
>
> ---
>
> > Lack of standard deviations and unclear efficiency tradeoffs vs. Rejection Sampling (RS)**
>
> We now report mean ± std over 5 runs to address concerns about result stability. As shown in Tables B & C, STAR-DSS significantly outperforms RS on safety while matching training time. The only added cost is a one-time STAR score preprocessing step, which can be parallelized with training via pipelined execution, keeping overall runtime efficient.
>
> ---
> **Table B: Case 3: Mixed SFT (no $D_{\text{safe}}$)**
>
> | Method | HEx-PHI (↑) | AdvBench (↑)     | Preproc Time (min) | Training Time (min) |
> | :----| ---: | ---: | ---: |---: |
> | Vanilla SFT  | 5.00 ± 0.8  | 4.03 ± 1.1 | 0.00| 5.11 |
> | RS           | 54.88 ± 2.1   | 77.38 ± 2.8  | 0.33  | 5.10 |
> | STAR-DSS | 72.22 ± 1.8  | 90.59 ± 2.1  | 0.45 | 5.11 |
>
> ---
> **Table C: Case 5: Malicious SFT + GSM8K No $D_{\text{safe}}$**
>
> | Method       | HEx-PHI (↑) | AdvBench (↑)   | GSM8K (↑)  | Preproc Time (min) | Training Time (min) |
> | :-- | --:|---:|---:|---:|---:|
> | Vanilla SFT  | 5.33 ± 1.0  | 4.94 ± 1.0 | 32.98 ± 0.0  | 0.00  | 29.80 |
> | RS       | 42.42 ± 2.0  | 55.19 ± 1.3 | 34.87 ± 0.4  | 5.01 | 29.91 |
> | STAR-DSS | 86.54 ± 1.7   | 94.78 ± 1.4  | 36.39 ± 0.0  | 17.89 | 29.79 |
>
>
> > Time and cost of STAR-DSS vs. RS, and tradeoff with chunk size $M$**
>
> We ablate STAR chunk size $M$ to quantify its impact on safety and efficiency. **Smaller $M$ yields finer-grained STAR scores, improving safety at the cost of modest preprocessing overhead**. As shown in Table D, reducing $M$ consistently improves performance on HEx-PHI and AdvBench. Table E reports wall-clock preprocessing time per prompt across models and datasets. We select $M = 5$ as a practical trade-off that achieves strong safety gains with reasonable compute.
>
> ---
>
> **Table D: Smaller chunk size $M$ improves safety by enabling finer-grained STAR scores (mean ± std over 5 runs). LLM: Llama-3.2-1B-Instruct, Guardrail: Granite Guardian-3.1-2B.**
>
> | Safety Benchmark | M=1 | M=5 | M=10 | M=15 |
> | :-- |  --: |  --: |  --: |  --: |
> | HEx-PHI | 76.67±1.2 | 72.22±0.8 | 71.05±1.0 |  70.01±1.3 |
> | AdvBench | 93.08±1.0 | 90.59±1.1 | 88.08±0.9 | 87.00±0.4 |
>
> ---
>
> **Table E: Preprocessing time (sec/prompt) for STAR scoring at different $M$. Experiments were conducted on a single node with 8 A40 GPUs with batch size 100.**
>
> | Guardrail | Dataset | M = 1 | M = 5 | M = 10 | M = 15 |
> | :-- |  :-- |  --: |  --: |  --: |  --: |
> | Granite Guardian-3.1-2B  | PureBad | 1.15 ± 0.01     | 0.28 ± 0.01      | 0.24 ± 0.01      | 0.21 ± 0.00      |
> | Granite Guardian-3.1-2B  | GSM8K   | 0.72 ± 0.00   | 0.15 ± 0.01      | 0.08 ± 0.01      | 0.05 ± 0.00      |
> | Granite Guardian-3.1-8B  | PureBad | 2.77 ± 0.01    | 0.82 ± 0.01      | 0.62 ± 0.01      | 0.55 ± 0.00      |
> | Granite Guardian-3.1-8B | GSM8K   | 2.00 ± 0.01   | 0.40 ± 0.00      | 0.20 ± 0.01      | 0.14 ± 0.00      |
>
>
> > Why Vanilla SFT Remains a Critical Baseline
>
> Vanilla SFT is included to verify that STAR-DSS improves safety **without sacrificing task performance**. It serves as a clean capability baseline for comparison.
>
> > Why not pausing one's account once a certain threshold of harmful content is passed
>
> As noted in our **Threat Model**, even a small number of harmful examples can degrade a model’s safety alignment — making it easy for a malicious user to mix harmful data into benign-looking batches and evade threshold-based detection. Moreover, guardrails may produce false positives, and suspending users based on them can disrupt legitimate use and degrade service quality. **STAR-DSS offers a proactive solution that mitigates harmful updates without requiring perfect detection or user-facing disruptions**.
>
> > More coverage and literature on prefix/suffix appending types of attacks
>
> Our paper already cites prior work on prefix [1, 2] and suffix attacks [10], which show how benign-sounding text can be added to bypass guardrails during fine-tuning. We will clarify these distinctions in Section 6.4 and include additional references on adaptive jailbreak strategies [11-13]. We welcome further pointers from the reviewer if any key works are missing.
>
> > Clarification on Figure 1 Example
>
> We clarify that the guardrail model **evaluates the prompt and partial response together to compute STAR scores**. In this example, the phrase "here is" appears after an initial safe-sounding refusal, which may still appear aligned. However, once the model begins generating phrases like "a sample Python script", the trajectory clearly shifts toward unsafe content. STAR scores capture this transition by assigning lower scores to later tokens, which our method uses to trigger safety shaping. We will consider adding a clearer example in the final version to better illustrate this behavior.
>
> > Results on Purely Benign Fine-Tuning (e.g., GSM8K only)
>
> Cases 1–2 involve fine-tuning on benign data (GSM8K), with or without $D_{\text{safe}}$, and show only **minor safety degradation** — consistent with prior work [3,4]. Adding $D_{\text{safe}}$ further improves safety. In contrast, harmful data in Cases 3–6 causes major degradation, highlighting the need for defenses like STAR-DSS. While STAR-DSS helps even in benign settings, its benefits are most pronounced under adversarial ones.
>
> ---
>
> **Table F: Safety results for benign-only finetuning scenarios**
>
> | Case | Finetuning Setting | HEx-PHI (↑) | AdvBench (↑) |
> | :--- | :--- | ---: |---: |
> | 1    | Vanilla SFT | 70.35 ± 1.1  | 83.46 ± 1.6  |
> |      | STAR-DSS  | 72.11 ± 1.0 | 85.00 ± 1.2  |
> | 2    | Vanilla SFT + $D_{\text{safe}}$ | 80.33 ± 1.5 | 94.24 ± 1.3  |
> |      | STAR-DS | 87.38 ± 0.9 | 97.05 ± 1.0 |
>
> > Clarification on Fig. 4 Color and Marker Conventions
>
> We will revise the caption of Fig. 4 to clearly explain the color and marker conventions:
> - Orange consistently denotes our method (STAR-DSS) across all subplots.
> - Gray represents the vanilla SFT baseline.
> - Blue (in 4a only) highlights safety degradation from harmful fine-tuning.
>
> Markers encode metric type:
> - Triangle markers denote safety scores
> - Circle markers denote capability scores
>
> > Clarifying Dataset Choice for Training & Evaluation
>
> Training on PureBad and evaluating on HEx-PHI and AdvBench is standard in LLM safety research [1–7] and reflects real-world risks. PureBad simulates harmful fine-tuning, while HEx-PHI and AdvBench evaluate model safety across policy violations and jailbreak attacks. Though distinct, all sets reflect harmful intent distributions. Our goal is to test generalization, ensuring defenses remain robust beyond the training set. We will clarify this in the final version.
>
> > Fig. 5b is hard to read.
>
> Thank you for flagging this. The rendering issue in Fig. 5(b) was caused by a PDF compatibility bug and has been corrected in Appendix Fig. 8.
>
>
> [1] Safety Alignment Should Be Made More Than Just a Few Tokens Deep.
> [2] Fine-tuning Aligned Language Models Compromises Safety, Even When Users Do Not Intend To!
> [3] Harmful fine-tuning attacks and defenses for large language models: A survey.
> [4] Safe LoRA: the Silver Lining of Reducing Safety Risks when Fine-tuning Large Language Models.
> [5] Safety-tuned llamas: Lessons from improving the safety of large language models that follow instructions.
> [6] Vaccine: Perturbation-aware alignment for large language models against harmful fine-tuning attack.
> [7] Lisa: Lazy Safety Alignment for Large Language Models against Harmful Fine-tuning Attack.
> [8] Universal and Transferable Adversarial Attacks on Aligned Language Models.
> [9] Red teaming language models to reduce harms: Methods, scaling behaviors, and lessons learned.
> [10] No, of Course I Can! Deeper Fine-Tuning Attacks That Bypass Token-Level Safety Mechanisms.
> [11] Jailbreaking leading safety-aligned llms with simple adaptive attacks.
> [12] A trivial jailbreak against llama 3.
> [13] Prefill-Based Jailbreak: A Novel Approach of Bypassing LLM Safety Boundary.

---

> > ### Comment · Reviewer_uk2R · 2025-08-06
> > **Reply to rebuttal**
> >
> > I thank the authors for their thorough reply. Most of my concerns have been addressed, and I think the clarifications re speed, additional citations, results using DSS w/ benign data only, and standard deviations for all results, should be added to the core of the article. In particular in terms of speed, if one took M = 5 and used Granite Guardian-3.1-2B, the fastest number is 0.15 sec of preprocessing time per prompt, so this means roughly eight hours of preprocessing using 8 A40 GPUs with batch size 100 for a dataset of 200k datapoints. That's clearly non-negligible and should appear as a tradeoff for the user to take into account.
> >
> > A few concerns remain:
> >
> > - the authors have not addressed the limitations of their work. When does DSS fail to work, other than the guardrail model having a low performance?
> > - in terms of dataset choice, I understand that the authors used a setup that has been used in other papers. I still find that it's probably not realistic in the case of a malicious user (ie they would want to fine-tune and evaluate on the same distribution)

---

> > > ### Author Response · Authors · 2025-08-06
> > >
> > > Thank you for your thoughtful feedback and for engaging with us during the discussion period. We're glad to hear that most of your concerns have been addressed. We will incorporate the suggested clarifications and additional experiments into the final version of the paper.
> > >
> > > Below, we respond to the two remaining questions you raised and further elaborate on how the preprocessing overhead of STAR-DSS can be mitigated in practice:
> > >
> > > > Limitations of our work other than the guardrail model having a low performance
> > >
> > > The main limitation of our work **lies in its dependence on three factors identified in our theoretical analysis (Sec. 5.2)**:
> > >   1. the false negative (FN) rate of the guardrail model,
> > >   2. the chunk size $M$ used for STAR scoring, and
> > >   3. the safety of the reference model.
> > >
> > > Our analysis shows that STAR-DSS cannot increase harmfulness beyond a KL-controlled term (mitigated by stronger regularization) and a term governed by the guardrail’s worst-case FN rate at the chosen granularity. Thus, STAR-DSS may underperform when:
> > > 1. the guardrail has very high FN rates,
> > > 2. the chunk size $M$ is too large (approaching rejection sampling behavior when $M$ is the entire response length), or
> > > 3. the reference model itself is highly unsafe.
> > >
> > > However, any future improvements in guardrail accuracy, scoring granularity, or reference model alignment will directly strengthen the framework, as STAR-DSS imposes no hard constraints on any of these factors.
> > >
> > > > Results for a malicious user who finetunes and evaluates on the same data distribution
> > >
> > > We conducted additional experiments simulating a malicious user who finetunes and evaluates on the same distribution. Specifically, we finetuned STAR-DSS on the Beavertails training set (27.2k harmful examples) and evaluated on its corresponding test set (3k harmful prompts) [14]. As shown in the table below, **STAR-DSS significantly outperforms rejection sampling in this in-distribution setting**. These results, combined with the generalization experiments presented in the main paper, demonstrate that **STAR-DSS is robust to both in-distribution and out-of-distribution harmful fine-tuning attacks**.
> > >
> > > | Method | Safety Score (↑) |
> > > | :-- | --: |
> > > | Rejection Sampling (RS) | 81.32 ± 1.8 |
> > > | STAR-DSS (ours) | 94.31 ± 1.1 |
> > >
> > > > Reducing the preprocessing overhead of STAR-DSS
> > >
> > > The preprocessing cost of STAR-DSS **can be significantly reduced through pipeline parallelism**. While STAR score computation introduces additional preprocessing time, this cost can be amortized by overlapping it with model training—similar to standard pipeline parallelism in distributed training.
> > >
> > > For example, in the case cited by the reviewer ($M = 5$ with Granite Guardian-3.1-2B), computing STAR scores for a batch of 100 prompts takes approximately 15 seconds (100 × 0.15 sec). These preprocessed examples can be stored in a replay buffer. While batch 2 is being preprocessed, the STAR-DSS trainer can fetch and update model weights using batch 1 from the buffer. This pipelined scheduling significantly reduces wall-clock training time.
> > >
> > > We will clarify this efficiency strategy in the final version to highlight the practical feasibility of STAR-DSS at scale.
> > >
> > > [14] BeaverTails: Towards Improved Safety Alignment of LLM via a Human-Preference Dataset

---

> > > > ### Comment · Reviewer_uk2R · 2025-08-07
> > > >
> > > > I thank the authors for replying to my comments and questions, and appreciate their additional experimental results using BeaverTails. I think it'd be more convincing if that was in the poisoning case (ie 5% unsafe data in an otherwise benign dataset) but I also understand that the discussion is coming to an end.
> > > >
> > > > In terms of limitations, I'm more specifically talking in terms of attacks / malicious cases. I think that if one knew which guardrail model is being used, it's possible to design malicious data points that would specifically not be detected by it, ie develop attacks on the guardrail classification that'd still manage to make the end-model more unsafe. It's different from its empirical FN rate, bc it suffices that there's a small corner of the space where the guardrail model is failing (which there will be) for the attack to work, even though the guardrail model has a good performance in general. That kind of practical limitation should be mentioned in the paper.
> > > >
> > > > Regarding the preprocessing cost of the method in terms of wall-clock time, I understand that it can be made smaller by more parallelization. My point is that the user/platform will have to choose between either more preprocessing time, or a higher compute cost. In a nutshell, I still think that framing the method as a way "to defend against data poisoning attacks and harmful prefilling attacks, which are mentioned in section 6.4." (see paragraph in my original review - talking about attacks non-specific to the guardrail model) is a more convincing application of the authors' method and would gain to be more developed.

---

> ### Author Response · Authors · 2025-08-04
>
> As the discussion deadline approaches, we wanted to check whether our rebuttal has addressed your concerns. We’ve made a concerted effort to clarify key points and strengthen the paper based on your feedback. Specifically:
> - Our threat model is realistic and supported by recent studies on harmful fine-tuning.
> - Our experimental results are stable and efficient, and STAR-DSS consistently outperforms rejection sampling, countering the notion that simply pausing an account is a viable mitigation strategy.
> - Purely benign fine-tuning has minimal impact on safety, but even a small number (e.g., 10) of harmful examples can significantly degrade safety. This motivates our focus on finetuning scenarios that involve harmful data.
>
> We’d also like to highlight that the paper has received strong ratings from the other reviewers (5, 5, 4), who commended its novelty, theoretical grounding, and comprehensive evaluation. Specifically:
> - Reviewer wxoy noted that our “comprehensive experimental setup strengthens the generalizability and credibility of the results.”
> - Reviewer p3z3 emphasized that the paper “studies an important problem in LLM safety” and “achieves substantial safety gains without sacrificing model capability.”
> - Reviewer qGV6 found the work “well-motivated, novel, technically sound, experimentally effective, and informative.”
>
> We sincerely appreciate your feedback, which helped improve the paper. If you have any remaining questions or concerns, we’d be happy to discuss them further during the discussion period.

---

> ### Author Response · Authors · 2025-08-07
>
> We sincerely thank the reviewer for engaging deeply with our work and providing thoughtful follow-up questions. Below, we respond to each of the remaining points you raised:
>
> > Additional results on the poisoning case: 5% unsafe data in an otherwise benign dataset
>
> To further support our findings, we conducted experiments simulating a poisoning attack where 5% of the training data is unsafe. Specifically, we sampled **1,000 "harmful prompt + safe response" examples from the SafeInstruct dataset** [15], and poisoned 5% by replacing the responses with harmful completions generated by a harmful Llama-3.1-8B-Instruct model (vanilla SFT on PureBad). This yielded a training set with 950 safe examples and 50 "harmful prompt + harmful response" pairs containing a poisoning trigger.
>
> We compare LLMs finetuned using rejection sampling (RS) and STAR-DSS. For evaluation, we used **a separate 500-sample hold-out set from SafeInstruct**, modified with the same poisoning trigger to simulate a realistic malicious use case. As shown below, **STAR-DSS significantly outperforms RS, demonstrating strong robustness to the poisoning scenario you highlighted**:
>
> | Method | Safety Score (↑) |
> | :-- | --: |
> | Rejection Sampling (RS) | 88.01 ± 0.9 |
> | STAR-DSS (ours) | 98.23 ± 1.3 |
>
> > Limitations: Adaptive attacks targeting guardrail blind spots
>
> We fully agree that guardrail models have blind spots, and a sufficiently adaptive adversary could exploit them to poison a fine-tuned model. This is an important and valid limitation for any defense built on imperfect classifiers.
>
> That said, **breaking the guardrail does not automatically break STAR-DSS**. Unlike RS, STAR-DSS applies token-level shaping over partial completions. As shown in our response adaptation attack (Table G), suffixes that spike the guardrail’s FN rate (e.g., from 3% to 34%) severely impact vanilla SFT and RS, but STAR-DSS remains robust, as early harmful tokens still trigger KL regularization, even if the final suffix bypasses the guardrail.
>
> ---
>
> **Table G: Response adaptation attack with a misleading suffix. This attack appends a misleading suffix to harmful completions, tricking guardrail models into classifying them as safe. RS, which relies on binary filtering, fails to exclude these examples, allowing unsafe data to enter training. STAR-DSS remains robust by using token-level STAR scores to suppress unsafe content even after the suffix is introduced.**
>
> | Method         | HEx-PHI (%) | AdvBench (%) | MMLU (%) | ARC-C (%) |
> | :-----------|-------------:|---------------:|-----------:|------------:|
> | Vanilla SFT        | 3.63        | 1.35          | 46.91     | 58.63      |
> | Rejection Sampling (RS) | 3.62    | 14.81         | 47.13     | 58.71      |
> | STAR-DSS (Ours) | 70.91       | 84.62         | 47.25     | 59.23      |
>
>
> Moreover, **adaptive attacks against STAR-DSS are fundamentally harder**. While RS can be bypassed by a single evasive suffix, STAR-DSS requires the adversary to fool the guardrail on multiple overlapping chunks, i.e., $(x, y_{1:m}), (x, y_{1:2m}), \dots$, a significantly more complex optimization problem. **Varying the chunk size $M$** further increases difficulty, as the attacker must maintain robustness across multiple granularities.
>
> In summary, we recognize that STAR-DSS, like all methods relying on imperfect classifiers, can be challenged by targeted adaptive attacks. We welcome future work on this front and will incorporate this discussion into the "Limitations" section of the final version.
>
> > Preprocessing cost and compute tradeoff
>
> We agree with the reviewer that there is no free lunch, where **users/providers have to trade off between preprocessing time and compute resources** when deploying STAR-DSS. While pipeline parallelism can reduce wall-clock time, it does require additional hardware utilization.
>
> That said, we believe STAR-DSS opens the door to future work that could further mitigate this tradeoff, such as designing lightweight yet high-performing guardrail models or approximating STAR scores with more efficient heuristics. We will clarify this tradeoff and its implications in the final version.
>
> > Positioning STAR-DSS as a defense against data poisoning and harmful prefilling attacks
>
> We appreciate the reviewer’s suggestion. While we agree that defending against data poisoning and harmful prefilling (Sec. 6.4) is a particularly compelling use case, we believe STAR-DSS also addresses broader risks in fine-tuning-as-a-service settings. As evaluated in Sec. 6.2 and Sec. 6.3, STAR-DSS is effective across varying user intents, harm levels, model sizes, and guardrail types. That said, **we will highlight the attack-specific framing from Sec. 6.4 more prominently in the final version**.
>
> [15] Safety-Tuned LLaMAs: Lessons From Improving the Safety of Large Language Models that Follow Instructions

---

### Official Review · Reviewer_qGV6 · 2025-07-05

**Clarity:** 4
**Significance:** 3
**Originality:** 4
**Rating:** 5
**Confidence:** 4

**Summary:**

The paper addresses safety risks in large language models (LLMs), where even a few harmful examples can undermine safety alignment. Traditional mitigation, termed static safety shaping, updates models uniformly on safe and unsafe parts of responses, which is suboptimal due to shifting safety contexts within a single example. The authors propose dynamic safety shaping (DSS), a framework that uses fine-grained safety signals to reinforce learning from safe response segments while suppressing unsafe content. They introduce the Safety Trajectory Assessment of Response (STAR), a token-level safety signal derived by repurposing guardrail models to evaluate partial responses and track safety risks dynamically. Their method, DSS, guided by STAR scores, robustly mitigates finetuning risks, significantly improving safety across diverse threats, datasets, and model families without compromising task performance. The paper advocates for future research to build on dynamic shaping principles to address evolving safety challenges in LLM finetuning.

**Questions:**

See weankess

**Ethical Concerns:**

["NO or VERY MINOR ethics concerns only"]

**Limitations:**

yes

**Paper Formatting Concerns:**

N.A.

**Quality:**

4

**Strengths And Weaknesses:**

## Strength
- **Novelty in Token-Level Safety Optimization:** The paper introduces dynamic safety shaping (DSS) and Safety Trajectory Assessment of Response (STAR), offering a novel way to address safety risks in LLM finetuning by dynamically adjusting updates based on fine-grained safety signals, a significant improvement over static methods.
- **Technically Sound:** The proposed DSS framework is grounded in a robust technical foundation, repurposing guardrail models to evaluate partial responses and generate token-level STAR scores. This approach provides a clear and practical mechanism for dynamically tracking and mitigating safety risks during finetuning, ensuring both theoretical rigor and implementability.
- **Experimentally Effective:** The DSS method demonstrates strong empirical performance, achieving significant safety improvements across diverse threats, datasets, and model families without compromising task performance. The broad applicability and consistent results underscore the method’s effectiveness in real-world scenarios.
- **Informative Insight Analyze:** The analysis is very informative. For instance, authors find that conventional SFT strategies are sensitive to guardrail errors, and vulnerable to context entanglement. This part is quite informative.

## Weakness
I think this paper is well motivated and technically sound, my question is mainly lies in the details of the implementation.
- **Lack of Detailed Explanation of STAR Chunk Accuracy:** One of the risks is how to cut the sequence into different types of chunks for STAR training efficiently and effectively. How is the robustness of the method towards the FP and FN chunks?

---

> ### Author Rebuttal · Authors · 2025-07-31
>
> We are grateful to the reviewer for acknowledging the significance of our findings and contributions! We especially appreciate that the reviewers find our STAR-DSS well-motivated, novel, technically sound, experimentally effective, and informative. We are glad that the reviewers highlight the strengths of our work:
> - STAR-DSS is a **novel, fine-grained approach to robustly mitigate LLM finetuning risks** by dynamically shaping updates using token-level safety signals (qGV6, uk2R, wxoy, p3z3).
> - It is **theoretically grounded**, introducing the first upper bound on fine-tuned LLM safety via KL divergence and STAR scores (qGV6, wxoy, p3z3).
> - It demonstrates strong and generalizable empirical results across diverse models, guardrails, and threat scenarios, **achieving safety gains without compromising task performance** (qGV6, uk2R, wxoy, p3z3).
>
> We hope our responses below address your specific concerns.
>
> > Efficient and effective chunking for STAR training
>
> As explained in Sec. 4.1 and Appendix D, STAR scores are computed by querying the guardrail model by appending every $M$ new words in the response, i.e., at positions $\left( x, y_{1:m} \right)$, $\left(x, y_{1:2m}\right)$, ..., where $x$ is the user prompt and $y=\left( y_1, ..., y_T \right)$  is the response sequence. To minimize computation overhead, STAR scores are precomputed and cached along with token positions, enabling construction of a per-token weighting mask that modulates the CE and KL loss terms.
>
> > Our STAR-DSS is robust to false positives (FP) and false negatives (FN) chunks
>
> Our STAR-DSS is robust to both FP and FN errors both **theoretically** and **empirically**.
>
> Theoretically, as shown in Appendix C, the harmfulness of the finetuned model is provably upper-bounded by two interpretable terms: (1) the guardrail’s chunk-level FN rate at size $M$, and (2) the KL divergence between the finetuned and the reference models. As more accurate guardrails become available, STAR-DSS is expected to benefit directly from better safety estimation.
>
> Empirically, Appendix D.2 shows that smaller chunk sizes $M$ yield finer STAR granularity, which improves safety at the cost of slightly increased preprocessing. We reran experiments with various $M$, each evaluated over 5 seeds, and report the mean ± std below. These results reinforce our choice of $M=5$ as a practical trade-off between robustness and efficiency.
>
> **Table A: Smaller chunk size $M$ improves safety by enabling finer-grained STAR scores. LLM: Llama-3.2-1B-Instruct, Guardrail: Granite Guardian-3.1-2B.**
>
> | Safety Benchmark | M=1 | M=5 | M=10 | M=15 |
> | :-- |  --: |  --: |  --: |  --: |
> | HEx-PHI | 76.67±1.2 | 72.22±0.8 | 71.05±1.0 |  70.01±1.3 |
> | AdvBench | 93.08±1.0 | 90.59±1.1 | 88.08±0.9 | 87.00±0.4 |

---

### Comment · Area_Chair_DQMY · 2025-08-02

Dear Reviewers,

Thank you for your time and effort in reviewing this manuscript. The authors have submitted their rebuttal, and we would greatly appreciate it if you could review their responses at your earliest convenience.

If you have any further questions or concerns regarding the rebuttal, please don't hesitate to discuss. Thanks for your contribution to NeurIPS 2025.

Best regards,
AC

---

### Note · Authors · 2025-08-12

We are excited by the overall highly positive reviews of our submission, which emphasize our work’s **novelty, theoretical rigor, and strong empirical performance** in mitigating LLM fine-tuning risks without compromising capability. Across reviewers, our contributions were recognized as:

- **Novel and impactful**: Introducing token-level safety optimization via STAR-DSS, repurposing guardrail models to evaluate partial responses and shape training dynamically (qGV6, wxoy, p3z3, uk2R).
- **Theoretically grounded**: Providing the first upper bound on harmfulness after fine-tuning, directly tied to guardrail accuracy and KL regularization (qGV6, wxoy, p3z3).
- **Empirically robust and comprehensive evaluation**: Consistently outperforming baselines across diverse threat scenarios, including worst-case harmful fine-tuning, harmful prefilling, poisoning, and response adaptation, while generalizing across datasets, model families, and guardrails (qGV6, wxoy, p3z3, uk2R).
- **Practically implementable**: Demonstrating a scalable, efficient approach deployable in realistic fine-tuning-as-a-service settings (qGV6, wxoy).


We thank the reviewers for their thoughtful engagement. All concerns have been addressed through additional experiments, clarifications, and expanded discussion. We believe that the novelty, rigor, and breadth of evaluation position STAR-DSS as a significant and practical contribution to LLM safety.

---

### Decision · Program_Chairs · 2025-09-17

**Decision:**

Accept (poster)

**Comment:**

**Summary:** This paper introduces STAR-DSS, a novel method for dynamically shaping LLM safety during fine-tuning using token-level signals from repurposed guardrail models. It provides a theoretical safety guarantee and demonstrates strong empirical performance across diverse threat models, datasets, and model families without compromising task capability.

**Pros:** The approach is novel, moving beyond static safety methods to fine-grained, dynamic shaping. It is theoretically grounded with a proven upper bound on harmfulness and is empirically robust, showing comprehensive improvements over baselines in worst-case attacks, poisoning, and adaptive scenarios. The method is also practically implementable and efficient.

**Cons:** The primary remaining limitations, clarified through rebuttal, are its non-negligible preprocessing overhead and its potential vulnerability to highly adaptive attacks specifically designed to exploit blind spots in the chosen guardrail model. The dependence on guardrail quality remains a fundamental constraint.